# High Mobility Group A (HMGA): Chromatin Nodes Controlled by a Knotty miRNA Network

**DOI:** 10.3390/ijms21030717

**Published:** 2020-01-22

**Authors:** Riccardo Sgarra, Silvia Pegoraro, Daniela D’Angelo, Gloria Ros, Rossella Zanin, Michela Sgubin, Sara Petrosino, Sabrina Battista, Guidalberto Manfioletti

**Affiliations:** 1Dipartimento di Scienze della Vita, Università degli Studi di Trieste, 34127 Trieste, Italy; spegoraro@units.it (S.P.); gros@units.it (G.R.); rzanin@units.it (R.Z.); MICHELA.SGUBIN@phd.units.it (M.S.);; 2Istituto per l’Endocrinologia e l’Oncologia Sperimentale (IEOS) “G. Salvatore”, Consiglio Nazionale delle Ricerche (CNR) c/o Dipartimento di Medicina Molecolare e Biotecnologie Mediche (DMMBM), Università degli Studi di Napoli “Federico II”, Via Pansini 5, 80131 Napoli, Italy; daniela.dangelo@unina.it

**Keywords:** High mobility group A, miRNA, post-transcriptional regulation, cancer

## Abstract

High mobility group A (HMGA) proteins are oncofoetal chromatin architectural factors that are widely involved in regulating gene expression. These proteins are unique, because they are highly expressed in embryonic and cancer cells, where they play a relevant role in cell proliferation, stemness, and the acquisition of aggressive tumour traits, i.e., motility, invasiveness, and metastatic properties. The HMGA protein expression levels and activities are controlled by a connected set of events at the transcriptional, post-transcriptional, and post-translational levels. In fact, microRNA (miRNA)-mediated RNA stability is the most-studied mechanism of HMGA protein expression modulation. In this review, we contribute to a comprehensive overview of HMGA-targeting miRNAs; we provide detailed information regarding HMGA gene structural organization and a comprehensive evaluation and description of HMGA-targeting miRNAs, while focusing on those that are widely involved in HMGA regulation; and, we aim to offer insights into HMGA-miRNA mutual cross-talk from a functional and cancer-related perspective, highlighting possible clinical implications.

## 1. Introduction

The high mobility group A (HMGA) family is a family of architectural nuclear proteins that are involved in the modulation of chromatin structure and regulation of gene expression. The family comprises three main members: HMGA1a and HMGA1b, which are derived from the alternative splicing of the *HMGA1* gene, and HMGA2, which is derived from a different—although related—gene [1,2]. These small proteins (approximately 12 kDa) contain three DNA-binding domains, the so-called “AT-hooks”, which allow for them to bind short A/T-rich sequences through the DNA minor groove, and a highly acidic C-terminal tail [3]. These proteins have high plasticity due to their intrinsic disordered structure, enabling their interaction with a multitude of factors [4,5]. The combination of these features allows HMGA factors to organize and orchestrate the assembly of stereospecific nucleoprotein complexes at the promoter/enhancer DNA sequence level [6], thus participating in regulating the expression of numerous genes [7,8,9]. In addition, HMGA proteins can participate in chromatin relaxation and the modulation of nuclear stiffness through mechanisms that involve histone H1 competition [10] and alterations in histone H1 post-translational modifications (PTMs) [11]. HMGA proteins are involved in several cellular processes, such as cellular proliferation [12], differentiation [13], senescence [14], apoptosis [15,16], inflammation [17], metabolism [18,19], autophagy [20], DNA replication [21], DNA repair [22,23], splicing [24], and viral integration [25], given their importance within the chromatin network.

HMGA proteins are highly and widely expressed during embryonic development [26,27], where they play essential functions, as demonstrated by individual and combined knockout (KO) of *Hmga1* and *Hmga2* [19,26,28,29]. Conversely, HMGA expression, particularly HMGA2 expression, is generally very low or absent in adult tissues. Therefore, establishing finely regulated control of HMGA expression is very important in the correct development and the maintenance of adult cellular homoeostasis. Accordingly, aberrant expression of HMGA proteins due to the dysregulation of their expression or the expression of mutated forms causes several diseases, such as different forms of neoplasia and metabolic disorders, which have been extensively reviewed [30,31,32], and it is involved in other pathologies such as polycystic ovary syndrome, sporadic Alzheimer’s disease, myocardial infarction, obesity, ischaemia, atherosclerosis, and sepsis [24,28,33,34,35,36,37]. Therefore, precise spatiotemporal regulation of the expression of HMGA factors is crucial in the correct development and preservation of adult physiological conditions. The transcriptional regulation of both the *HMGA1* and *HMGA2* genes has been extensively and recently reviewed: *HMGA1* is an inducible gene that is mainly regulated by transcription factors at different promoter regions and enhancers, while the *HMGA2* promoter seems to be constitutively active in different cell lines, and its activity can be modulated either positively or negatively by different DNA-binding factors [38]. In addition, an R-loop-based mechanism has been demonstrated to be involved in *HMGA2* gene transcription modulation, providing an open chromatin conformation for *HMGA2* transcriptional cis-regulatory sequences [39]. Moreover, HMGA2 protein levels can also be modulated by stabilizing interactions with long non-coding RNA (lncRNA) molecules [40].

HMGA proteins are subjected to many PTMs that regulate their ability to bind DNA and they interact with several other factors; therefore, post-translational regulation is a relevant step contributing to the regulation of their activity [41].

Post-transcriptional regulation is a key process that regulates gene expression, and it is often altered in cancer cells [42,43]. Although both the 5′ untranslated region (5’UTR) and 3′UTR can contribute to this process, the 3′UTR in particular is more often a target of microRNAs (miRNAs), which are the most important factors that are involved in this type of regulation. Hundreds of papers have described miRNA-mediated regulation of both HMGA1 and HMGA2 mRNA in different cell types and stages (Table 1); in particular, HMGA2 regulation by let-7 is considered to be a paradigm of a miRNA action [44,45,46]. Moreover, lncRNA provides another layer of post-transcriptional regulation complexity; lncRNAs have been demonstrated to play a role in modulating HMGA expression by sponging HMGA miRNAs [47,48].

A comprehensive overview and critical evaluation of the role of miRNA in regulating HMGA1 and/or HMGA2 protein expression is still missing from the literature, despite this knowledge. In this review, we specifically focus on the HMGA-targeting miRNA network. We (i) provide insights into the structural organization of *HMGA* genes; (ii) offer a complete overview of HMGA-targeting miRNAs, focusing on those that have been more deeply investigated; (iii) discuss the hypothesis of the mutual influence of HMGA1/HMGA2; and, (iv) highlight the role of HMGA mRNAs as competing endogenous RNA (ceRNA) molecules in the context of cancer initiation and development.

## 2. *HMGA* Genes: Structural Organization

Analysis of the organization of the *HMGA1* gene [1,178] identified eight exons, several promoter regions, several transcription start sites, and numerous alternatively spliced exons, which generate different mRNAs encoding the two major protein isoforms (HMGA1a and HMGA1b) expressed in human cells. Interestingly, the three DNA-binding domains of the HMGA proteins, the AT-hooks, are encoded by three different exons, while the last exon encodes the acidic C-terminal tail and it includes the 3′UTR. The referenced papers and other papers [179,180,181] indicate that the 5′UTR of HMGA1 can be very heterogeneous, due to the use of different promoters and the occurrence of alternative splicing within the first four exons. However, no variants were described in the last four exons, which include the open reading frames (ORFs) and 3′UTR, except for the alternative splicing event in the first of these exons that generates either HMGA1a (HMGA1-201; HMGA1-205) or HMGA1b (HMGA1-204; HMGA1-202; HMGA1-203) (Figure 1) and the alternative splicing event that results from the usage of a non-canonical splice site that generates the rare HMGA1c transcript, which has only been described once and it is not present in Ensembl Genome Browser [182]. Remarkably, all of the transcripts that are described in these papers share the same 3′UTR.

Currently, whole-genome sequencing projects, together with transcriptome analysis, can provide a more complete picture of transcripts that originate from a specific locus. Starting from this consideration, we investigated whether new coding transcripts can arise from the human *HMGA1* gene. We used Ensembl Genome Browser 96 for this purpose, which collects open-access, integrated genome, gene, variation, gene regulation, and comparative genomic information [183] (Figure 1). This analysis did not reveal further complexity in HMGA1 transcripts; therefore, we can conclude that all of the transcripts share the same 3′UTR and can thus be targeted by the same miRNAs.

The *HMGA2* gene immediately appeared to be more complex than the *HMGA1* gene, essentially due to its length and the presence of a very large intron involved in rearrangements, leading to truncated or chimeric HMGA2 transcripts especially in benign mesenchymal tumours [184,185]. Indeed, the *HMGA2* gene was identified, because it was found to be rearranged in several tumours at the level of the third intron [184,185] and was only subsequently completely cloned and characterized [2,186]. The *HMGA2* gene is organized into five exons that encode different protein domains. Unlike *HMGA1*, it contains an additional exon coding for a very short peptide of 11 amino acids (aa) that separates the final DNA-binding domain from the acidic C-terminal tail. Within the long intron, some additional exons have been described to be part of alternative splicing transcripts [187,188]. We performed analysis on publicly available transcriptomic data to better define the HMGA2 transcripts originating from the *HMGA2* locus (Figure 2). In addition to the canonical HMGA2 transcript (HMGA2-204, according to Ensembl nomenclature), six splicing variants were identified, each of which ends with a different 3′UTR sequence and encodes a protein with a different C-terminal tail. Four of these variants (HMGA2-201; HMGA2-206; HMGA2-205; and, HMGA2-203) contain exons that are derived from the large third intron, while two (HMGA2-210 and HMGA2-202) contain exons derived from the fourth intron. In conclusion, unlike *HMGA1*, which only includes one 3′UTR, *HMGA2* has several splicing variants with 3′UTRs different from the canonical 3′UTR. This difference could have implications for HMGA2-mediated miRNA regulation. In fact, it has been reported that an alternative isoform that escapes miRNA-mediated targeting (HMGA2-203) is involved in human haematopoietic stem cell development [189].

## 3. miRNA Regulation of HMGA Transcripts

More than one hundred miRNAs are involved in the regulation of HMGA mRNA (Table 1). Among these miRNAs, we discuss, in detail those that are most widely studied, beginning with the let-7 family, which is one of the first examples of miRNA-mediated oncogene regulation [44,45].

### 3.1. The lin28/let–7 Axis

The description of miRNA-mediated HMGA protein regulation begins with the finding that chromosomal abnormalities in the regions 12q15 and 6p21.3 were associated with the aberrant expression of either HMGA2 or HMGA1 in several benign mesenchymal tumours [190]. Invariably, in the aberrant HMGA transcripts, the region coding for the N terminal had intact DNA-binding domains, while the region coding for the C-terminal tail, which included the 3′UTR, was deleted or substituted with other transcripts. Together with the frequent finding of a lack of correspondence between HMGA protein and mRNA levels, particularly for HMGA2, these data led to the hypothesis that regulatory elements within the 3′UTR could mediate the post-transcriptional control of HMGA protein expression. The first clue was provided by the finding that sequences in the 3′UTR of both HMGA1 and HMGA2 mRNAs could negatively control the expression of these mRNAs, whereas the deletion of these sequences—mimicking cytogenetic aberrations found in human tumours—led to protein overexpression [191]. The discovery that miRNAs of the let-7 family, which are potent regulators of larval development and adult fate specification in nematodes [192], are also expressed in human cells [193] paved the way for completing the puzzle: these ancient small molecules could also target six sites in the 3′UTR of HMGA2 [44,45], which represses its expression via mRNA degradation. The final picture was completed by the observation that translocations of the *HMGA2* gene leading to the loss of 3′UTR-mediated let-7 repression could activate HMGA2 expression and lead to cell transformation, anchorage-independent growth [44], and proliferation [45]. Following translocation, the HMGA2 3′UTR was frequently found to be bound to the 3′ end of tumour suppressor genes, such as RAD51L1 and FHIT [44], thus further promoting tumorigenesis.

Unlike in *C. elegans*, the human let-7 miRNA family includes 13 evolutionarily conserved members (*let-7a-1*, *7a-2*, *7a-3*, *7b*, *7c*, *7d*, *7e*, *7-f1*, *7f-2*, *7g*, *7i*, *mir-98*, and *miR-202*) [194,195] that share the same seed sequence, but are located on eight different chromosomes [196]. In addition to HMGA2 truncation, a decrease in let-7 expression can be solely responsible for the increased expression of otherwise normal HMGA2 or HMGA1 [51] in several different cancers, such as breast [69], gastric [70] and non-small cell lung cancers [63], sarcomas [197,198], hepatocellular carcinomas, nasopharyngeal [72] and oesophageal squamous cell carcinomas [199], uterine leiomyomas and leiomyosarcomas [66,200], and pituitary adenomas [201], as a loss of let-7 expression is a marker for less well-differentiated cancers [46]. Moreover, a role for let-7/HMGA in epithelial-mesenchymal transition (EMT), cell migration, and metastasis has been demonstrated in several cancers and cellular systems [51,72,202,203,204,205].

In addition to its role in cancer, the let-7/HMGA2 axis was shown to regulate physiological processes, such as adipose [206], osteogenic [71], myeloerythroid [207] and gliogenic differentiation [208], post-natal proliferation and ageing [209], and glucose metabolism [210]. This role mechanistically explains the finding of let-7 downregulation in pathologies, such as diffuse lipomatosis [211] and renal fibrogenesis [212]. The let-7/HMGA2 axis affects self-renewal and stemness in stem cells of different origins, including haematopoietic, neural, breast, lung, and intestinal cancer stem cells ([69,213,214,215,216], respectively). Most importantly, the let-7b/HMGA2 axis has been shown to induce the direct conversion of adult somatic cells into induced neural stem cells [56].

The factors controlling let-7 expression form a regulatory feedback loop with downstream targets of let-7 and are pivotal regulators of stemness or differentiation. The major molecules regulating let-7 are the two RNA-binding proteins LIN28A and LIN28B, [217], which recruit terminal uridyltransferase (TUT4) and add an oligomeric U at the 3′ end, which prevents pre-let-7 from being processed into a mature miRNA [218]. Strikingly, the LIN28 proteins dramatically affect HMGA1 and HMGA2 through both let-7-dependent and let-7-independent mechanisms [195]. The components of this network are linked by entangled positive and negative reciprocal regulatory activities that each factor exerts on the other factors and on itself (Figure 3) [219].

### 3.2. hsa-miR–26a

The hsa-miR-26 family includes miR-26a and miR-26b. miR-26a is specifically located on chromosome 3p22, a region that is subjected to the loss of heterozygosity in cancer [220]. In the physiological context, miR-26a controls cell growth, development, and differentiation in processes, such as myogenesis [221]. Via the transfection of cells with vectors containing a wild-type or mutated 3′UTR of HMGA1 or HMGA2 and reporter gene assays, several studies have demonstrated that miR-26a specifically targets well-conserved regions of the 3′UTRs of HMGA1 [87,89,90,92,95] and HMGA2 [91,93,94]. miR-26a downregulates HMGA1 at both the mRNA and protein levels, as assessed by qRT-PCR and Western blot analyses [87,90,92], while HMGA2 downregulation has only been validated at the protein level [93,222]. The role of the miR-26a/HMGA1 axis in the context of cancer has been extensively studied, and this axis has been shown to act on the proliferation and migration of pancreatic cancer [96], bladder cancer [95], breast cancer [92], lung adenocarcinoma [90], and osteosarcoma [89] cells. A negative correlation between miR-26a and HMGA1 levels has also been shown in cancer tissue specimens, specifically in tissues from urothelial bladder cancer patients [223] and osteosarcoma patients- [89]. Moreover, miR-26a acts on the inflammatory process by downregulating HMGA1 and MALT1, thus impacting the TNF-α/NF-κB inflammatory genes in human bronchial epithelial cells [88]; in addition, the miR-26a/HMGA1 axis controls coronary microembolization-induced myocardial inflammation [87]. miR-26a target HMGA2 mRNA, which regulates cellular senescence [224] and EMT in idiopathic pulmonary fibrosis both in vitro and in vivo [93]. In addition, miR-26a and HMGA2 mRNA are inversely correlated in gallbladder cancer (GBC) tissues, and miR-26a reduces the proliferation of GBC cells [94] and human lung adenocarcinoma cells, which increases the sensitivity to cisplatin treatment via HMGA2 [91].

### 3.3. hsa-miR–33b

The intronic human hsa-miR-33b-5p belongs to the *miR33* gene family and it is located in a non-coding region of the human *SREBP-1* gene at the 17p11.2 genomic locus (miRbase, HGNC) [225]. miR-33b can target the HMGA2 3′UTR, which prevents the expression of the HMGA2 protein [101,102,103,104].

miR-33b is downregulated in breast tumour and melanoma tissue samples and it is inversely correlated with the clinical stage [101,102].

Ectopic expression of miR-33b, which affects the levels of HMGA2, SALL4 and Twist1, can decrease the stem cell-like properties of breast cancer cells. Moreover, miR-33b suppresses cell migration and invasion in vitro through decreases in HMGA2 and Twist1 expression [101]. miR-33b also reduces the migration and invasiveness of melanoma cell lines upon cordycepin exposure via targeting the HMGA2 3′UTR [102]. miR-33b exerts its anti-tumorigenic effects on human gastric cancer cells by affecting HMGA2 expression to inhibit cell growth and increase cellular sensitivity to docetaxel and cisplatin [104]. In addition, the upregulation of miR-33b by EF24, which is a curcumin analogue, suppresses EMT and the induction of migration in melanoma cell lines via the targeting of HMGA2 [103].

miR-33b modulates pathways controlling the levels of high-density lipoprotein (HDL) cholesterol, triglycerides, and insulin signalling, and negatively regulates adipogenesis [100,226]. miR-33b expression was found to be elevated in ovarian tissues of rats with insulin-resistant polycystic ovary syndrome and to inhibit GLUT4 expression by targeting HMGA2 3′UTR, therefore contributing to the progression of insulin resistance in PCOS/IR rats [80]. Furthermore, HMGA2 mediates the effects of miR-33b on adipogenesis in the cells from patients with Simpson-Golabi-Behmel syndrome [100].

### 3.4. hsa-miR–98

*hsa-miR-98-5p* belongs to the *let-7* gene family and it is located at the Xp11.22 genomic locus (miRbase, HGNC site) [225]. Several studies have reported that miR-98 binds the 3′UTR of HMGA2 mRNA, regulating its expression [107,108,109,110,111,112]. A correlation between miR-98 and chemoresistance that is associated with the downregulation of HMGA2 was found in head and neck squamous cell carcinoma, providing the first link between miR-98 levels and HMGA2 expression [227]. Other authors have reported that miR-98 reduces tumour aggressiveness through the inhibition of HMGA2 expression [107,108,110,111]. miR-98 was shown to downregulate HMGA2 expression in glioma cells as a part of the RKIP/miR-98/HMGA2 axis, leading to a reduction in glioma cell invasion [110]. In breast cancer, retinoblastoma, and laryngeal squamous cell carcinoma, restoration of miR-98 with the consequent inhibition of HMGA2 expression reduced the proliferative and migratory abilities of cells, EMT, and metastasis formation in vivo [107,108,111]. Another work showed that miR-98 could reduce the inflammatory response and alleviate neuropathic pain progression in chronic constriction injury-induced (CCI) rats by binding to the HMGA2 3′UTR, [112]. The miR-98/HMGA2 axis is also involved in differentiation, as the expression of miR-98, with the consequent downregulation of HMGA2, promotes the expression of the osteogenic differentiation genes RUX2, BSC, and OCN in mesenchymal stem cells [109].

### 3.5. hsa-miR-16

miR-16 is considered to be a critical tumour suppressor miRNA downregulated in many types of cancer, among the validated miRNAs targeting both HMGA1 and HMGA2 [228]. Via a combined bioinformatic and molecular approach, HMGA1 was demonstrated to be directly targeted by miR-16 [84]. A luciferase assay that was performed on HMGA1 mutant constructs at predicted miR-16 binding sites determined that only one site is active. Moreover, the overexpression of miR-16 induced a decrease in the endogenous HMGA1 protein level due in part to a decrease in the HMGA1 mRNA level. In fact, a decrease in HMGA1 mRNA has only been found in HeLa cells, but not in MCF-7 cells, revealing a cell-dependent mechanism for HMGA1 regulation [84]. Later, another group confirmed these results on HMGA1 and demonstrated that miR-16 could also directly target HMGA2 [50]. Bioinformatic tools predicted one miR-16 binding site in the 3′UTR of HMGA2; subsequently, via a luciferase assay, miR-16 was demonstrated to target HMGA2. Consistent with previous observations, the overexpression of miR-16 reduced both HMGA1 and HMGA2 mRNA and protein levels in the GH3 rat pituitary adenoma cell line. Accordingly, another work described the relationship between miR-16 and HMGA2 in pituitary adenoma cells [83]. Interestingly, both works showed an inverse association between miR-16 and HMGA expression levels in human pituitary adenomas; in fact, they found that miR-16 was downregulated, but HMGA1 and HMGA2 were upregulated in pituitary adenomas when compared to normal pituitary tissues, consistent with the action of miR-16 in regulating the expression of HMGA1 and HMGA2 [50,83]. Recently, a new layer of miR-16/HMGA2 regulation in pituitary tumours has been added—ribosomal protein SA pseudogene 52 (RPSAP52), an antisense lncRNA targeting the *HMGA2* gene was demonstrated to increase HMGA2 protein expression via a ceRNA mechanism, in which it acts as a sponge for miR-16 and miR-15 [48].

### 3.6. hsa-miR-142-3p

hsa-miR-142 shows pleiotropic functions in physiological processes that are connected with embryonic development and stem cell pluripotency, whereas the alteration of its expression supports the development of cardiovascular disease [229] and tumours [133,230,231,232]. Decreased levels of miR-142-3p have often been reported in cancer tissues when compared with normal tissues, supporting the data while considering miR-142-3p mainly an onco-suppressor miRNA [233,234,235]. Interestingly, the “guide strand” of miR-142-3p is negatively modulated by interleukin-6 in glioblastoma multiforme (GBM), the most aggressive and stem cell-rich primary brain tumour [133], causing the upregulation of HMGA2 protein expression. Indeed, miR-142-3p decreased HMGA2 protein levels. This decrease led to inhibited expression of *SOX2*, which is a target gene of HMGA2 and a master regulator of stemness features, thus inducing the suppression of cancer cell pluripotency and tumour cell growth. Therefore, in GBM, miR-142-3p downregulation is required to maintain stem cell-like properties and cell proliferation through HMGA2 upregulation. Metformin, which is a first-line drug for type 2 diabetes mellitus, was proposed to disrupt the sponge effect of MALAT1 on miR-142-3p, enabling miR-142-3p to bind the HMGA2 3′UTR in cervical cancer and inhibit cervical cell invasion and migration [132]. Moreover, the miR-142-3p response element is also shared by HMGA1. Indeed, an inverse correlation between miR-142-3p and HMGA1 expression levels was found in osteosarcoma tissues, and miR-142-3p was subsequently demonstrated to inhibit the growth, migration, and invasion of osteosarcoma cells by targeting HMGA1 [134].

## 4. HMGA1 and HMGA2: Common and Specific miRNAs

Among the 75 miRNAs experimentally explored for their ability to target HMGA mRNA, 26 targeted HMGA1, 64 targeted HMGA2, and 15 targeted both HMGA1 and HMGA2 (Table 1). We used TargetScan software [236] to identify predicted HMGA-targeting miRNAs and compare them with those that have been experimentally validated. We arbitrarily restricted the list of predicted HMGA-targeting miRNAs to consider both miRNA families broadly conserved among vertebrates and miRNA families conserved only among mammals. A total of 132 predicted HMGA-targeting miRNAs were identified; 41 targeted HMGA1, 115 targeted HMGA2, and 24 targeted both HMGA1 and HMGA2 (Table 1). The number of predicted miRNAs is approximately twice that of experimentally validated miRNAs; however, notably, the target distribution in the two categories is almost identical (Figure 4, panel A). The experimental and bioinformatic data both suggest that the modulation of HMGA2 protein expression is controlled by a more wide miRNA network than the modulation of HMGA1 protein expression, which could simply be because the HMGA2 3′UTR is longer than the HMGA1 3′UTR (Figure 2).

A comparison between experimentally validated and bioinformatically predicted miRNAs revealed that of the experimentally validated miRNAs targeting HMGA1 (26), 54% (14) were also predicted by TargetScan and that almost the same percentage (61%; 39 miRNAs) of the 64 miRNAs targeting HMGA2 were also predicted by TargetScan (Figure 4, panel B). miRNA target prediction programs usually generate many false positives [237]. However, while only focusing on miRNAs that have been experimentally validated, it is evident that a very high number of miRNAs are involved in modulating HMGA protein expression. In general, the quantitative effect of miRNA target repression is very limited [238,239]; therefore, it is reasonable to hypothesize that the HMGA protein expression level could be controlled by a set of cooperating miRNAs.

For each predicted miRNA, we report in Table 1 the cumulative weighted context ++ score (CWCS), which is a score that is related to the predicted efficacy of the sites. On the basis of this score, this table allowed for us to speculate that the miRNA network that is involved in regulating HMGA protein expression could be even more connected than it actually is and some miRNAs could very possibly be added to the experimentally validated list. A set of miRNAs already experimentally validated to target one HMGA mRNA could be reasonably tested for targeting of the other HMGA mRNA: miR-98-5p and miR-107 could be validated for HMGA1, miR-424-5p could be another miRNA targeting both HMGA mRNAs, and miR-497-5p could be validated for HMGA2. Moreover, a set of miRNAs with promising CWCSs (i.e., miR-17-5p, miR-20-5p, miR-93-5p, miR-132-3p, miR-137, miR-190-5p, miR-212-3p, miR-212-3p, miR-519-3p, miR-532-3p, miR-485-5p, miR-491-5p, and miR-760) could be reasonably tested.

We believe that the shared regulation of HMGA1 and HMGA2 by a set of miRNAs implies a degree of interdependence between HMGA1 and HMGA2. Studies have already demonstrated that the expression of HMGA2 could be controlled by the expression levels of the HIF1A 3′UTR via a ceRNA mechanism [240]. Therefore, a finding that the expression of the two *HMGA* genes might be linked through a ceRNA mechanism would not be surprising. A bioinformatic search for mRNAs, which can act as ceRNAs that target HMGA1 or HMGA2, showed that HMGA2 mRNA had the highest ceRNA score for HMGA1 (Competing Endogenous mRNA DataBase (ceRDB), https://www.oncomir.umn.edu/cefinder/index.php). [241]. Let-7 is a master regulator of HMGA2, and the HMGA2 3′UTR contains 7 predicted let-7 consensus sequences, while the HMGA1 3′UTR contains only one. Simple reasoning suggests that, at least regarding the effects of let-7, the HMGA2 3′UTR-sequestering activity is very high and, therefore, even low HMGA2 expression levels could have a relevant impact on HMGA1 expression. In contrast, only a very high expression level of HMGA1 could relieve the suppressive effects of let-7 on HMGA2. The targeting of both mRNAs by multiple miRNAs suggests a degree of interdependence between the two proteins; however, in most research that is related to HMGA proteins, only one protein (i.e., HMGA1 or HMGA2) is studied without considering the possible interdependence with the other. This aspect might be considered in future studies.

While considering that the transcription of the two *HMGA* genes is controlled by different pathways/stimuli [38], the post-transcriptional modulation via miRNA adds another layer to the regulation of HMGA protein expression, and the integration of transcriptional and post-transcriptional mechanisms could dictate the relative HMGA1/HMGA2 protein expression levels.

## 5. HMGA–Targeting miRNAs: Their Role in the Modulation of Other Relevant Pathways Involved in Cancer

The power of miRNA regulation depends, in part, on the ability of these molecules to simultaneously target several mRNAs, thus affecting different pathways and inducing a fan-shaped effect. The types of targets and the pathways affected, however, occasionally make defining a miRNA as an oncogene or onco-suppressor difficult, as one miRNA sometimes performs opposing functions. The existence of self-regulating feedback loops or the effect of different cellular backgrounds might explain these dichotomies.

An analysis of the miRTarBase database (http://mirtarbase.mbc.nctu.edu.tw) [242] allowed for us to highlight additional and strongly validated targets (assessed by reporter assays, Western blotting and qPCR) of the major HMGA-targeting miRNAs. The let-7 family, in particular, is a miRNA family with several validated targets, including cell cycle regulators (such as MYC, HRAS, KRAS, CDC25A, NKIRAS2, CDC34, TGFBR1, BLIMP1 (or PRDM1), CCND1, CCND2, CDKN2A, and E2F2), transcriptional and translational regulators (such as NF2 and IGF2BP1), apoptosis regulators (BCL2L1), stem cell regulators (TRIM71), and EMT mediators (ITGB3). A potent positive feedback loop links inflammation and cancer through the NF-κB/LIN28/let-7 axis. NF-κB/LIN28-mediated let-7 repression leads to the activation of the IL-6, STAT3, and NF-κB mediators (Figure 3), thus maintaining the epigenetic transformed state in the absence of the inducing signal [243].

On the other hand, some let-7 targets function as molecular sponges that can remove let-7 molecules from other targets, hence functioning as oncogenes. In MYCN-amplified neuroblastoma cell lines, MYCN mRNA levels are exceptionally high and they are sufficient for sponge let-7 [244]. Not only mRNAs but also lncRNAs, can sponge let-7 and regulate important physiological processes, as seen for the lncRNA H19, which harbours several let-7 binding sites and mediates muscle differentiation [245].

MirTarBase screening for strongly validated targets shows that miR-26a also affects tumorigenesis, by often acting as a tumour suppressor and targeting genes that regulate fundamental cancer hallmarks, including the histone methyltransferase EZH2 [246,247], the pro-metastatic Methaderin (MTDH) [248], and the cell cycle regulator CCNE2 [249]. miR-26a participates in differentiation and angiogenic programs by targeting SMAD1 [250], whereas, in human gliomas, it is amplified and can target PTEN [251] and RB1 [252].

Validated, relevant miR-33b targets include PIM1 [253] and XIAP, both of which are involved in apoptosis [254], and ZEB1 [102,255], and the aforementioned TWIST1 [102], whose cooperation in the miR-33b/HMGA2/Twist1/ZEB1 axis plays an important role in modulating melanoma dissemination [102].

The strongly validated miR-98-5p targets include EZH2 [247], IL-6 [256], PGRMC1 [257], HK2 [258], ITGB3 [259], E2F2 [260], and FUS1 [261].

As miR-16 is considered to be a crucial onco-suppressor, many researchers have investigated the mechanism by which it affects cancer properties and have revealed that it is involved in targeting numerous genes that are crucial in mediating different cancer hallmarks [228]. Via analysis with the MiRTarBase tool, we found 34 strongly validated targets in addition to HMGA1 and HMGA2. Among these targets were CDK6, CCND1, CCND3, and CCNE1 [262], which indicated that miR-16 is involved in a mechanism by which it represses proliferation. Moreover, miR-16 targets VEGFA [234,263], YAP1 [264], FGF2, and FGFR1 [265], indicating its critical role in regulating the signalling, angiogenic, migratory, and invasive properties of cancer cells.

miR-142-3p has been validated to target a wide range of mRNAs [229]. Among these mRNAs, some factors that are crucial in cancer progression, specifically in metastasis promotion, such as TGFBR1 [266], RAC1 [233], and ROCK2 [267,268], were highlighted in the miRTarBase tool, emphasizing the role of miR-142-3p as an onco-suppressor.

## 6. HMGA–Related lncRNAs: Their Role as ceRNAs for HMGA-Targeting miRNAs

The abundance of miRNA response elements (MREs) in the HMGA1 and HMGA2 sequences not only makes them susceptible to negative regulation by miRNAs, but can also influence other transcripts or non-coding genes, such as lncRNAs or pseudogenes that share the MREs. Thus, the HMGA1 and HMGA2 mRNAs can reciprocally regulate each other by acting as ceRNAs.

One of the first lines of evidence indicating that HMGA mRNAs participate in the ceRNA network came from the identification and characterization of two HMGA1 non-coding processed pseudogenes, *HMGA1P6* and *HMGA1P7*. The conserved seed sequences in miRNAs that target the HMGA1 and HMGA2 mRNAs that are also present in the HMGA1 pseudogenes transcripts can protect HMGA mRNAs from miRNA inhibition; therefore, pseudogene overexpression can sustain cancer cell proliferation and migration and inhibit cell death [269]. Consistent with this finding, *HMGA1* pseudogenes upregulate several cancer-related genes [269], and RNA sequencing (RNA-Seq) of *HMGA1P7* transgenic mouse embryonic fibroblasts (MEFs) revealed that HMGA1P7 mRNA induces overexpression of H19 and Igf2 by acting as ceRNA [270]. Interestingly, the overexpression of miR-483 and miR-675, whose maturation is dependent on the transcription of H19 and Igf2, is dependent on early growth response protein 1 (Egr1), whose level is directly increased by HMGA1P7 via miRNA sponging. The consequent increase in the Egr1 level induces H19 and Igf2 transcript maturation and then generates miR-483 and miR-675, two oncogenic miRNAs (oncomiRs) that are overexpressed in several neoplasias [271].

Accumulating lines of evidence reveal that lncRNAs are also critical regulators of HMGA-dependent tumour growth. HOXC13-AS, a lncRNA that is involved in the HMGA2 ceRNA network, was recently identified. It can function as a ceRNA in nasopharyngeal carcinoma to promote the expression of HMGA2 by sponging miR-383-3p, and its silencing induced cell cycle arrest and apoptosis, consistent with the inhibition of the pro-proliferative function of HMGA2 [272].

Moreover, a very recent analysis of lncRNA expression in gonadotroph adenomas revealed the RPSAP52 gene, which is the head-to-head natural antisense transcript of *HMGA2*, among the most highly upregulated lncRNAs. RPSAP52 acts as an endogenous sponge, which protects HMGA1 and HMGA2 from miR-15a-, miR-15b-, and miR-16-mediated inhibition. RPSAP52 modulates HMGA expression and promotes cell proliferation by acting as a decoy for miRNAs and, most likely, redirecting them to other genes by interacting with RNA-binding proteins to alter the local DNA conformation [48]. Intriguingly, RPSAP52 can also enhance HMGA2-IGF2BP2-RAS axis activity and the balance between the LIN28B and let-7 levels [273] and can positively regulate *HMGA2* transcription [39]. Table 2 shows a list of lncRNAs that are involved in the HMGA-ceRNA network.

## 7. HMGA Post-Transcriptional Regulation: Is It Only a Matter of miRNA?

As the discovery of miRNA-mediated post-transcriptional regulation of RNA in *C. elegans* [192], this mechanism of regulating RNA stability and translation has been described in other organisms and it is now considered to be the most prevalent example of post-transcriptional regulation, at least in mammals. The observation that the 3′UTRs of mammalian RNAs are highly conserved [278] and extensively bound by proteins [279] suggests that other mechanisms, known in part but largely unexplored, could regulate RNA stability, translation, and sub-cellular localization. For example, the *Hmga2* transcript was selected as a case study, and regulatory sequences in its 3′UTR were systematically identified [280]. A high-resolution map of the 3′UTR was generated via reporter gene assays, and numerous previously unidentified regulatory sites that were well conserved across species were found. Interestingly, several sites with a positive effect on gene reporter expression have been discovered, including some candidate positive regulatory elements containing U-rich or CU-rich sequences consistent with AU-rich elements (AREs), in addition to previously identified let-7 miRNA target sites. Intriguingly, very recently, it was shown that the circular RNA *circ*NSUN2, by interacting with a short sequence within an ARE in the 3′UTR of HMGA2 mRNA, plays a critical role in promoting the formation of a ternary complex with the RNA-binding protein IGF2B2 that stabilizes HMGA2 mRNA to promote colorectal liver metastasis [281].

In addition, the RNA-binding protein IMP3 has been found to form cytoplasmic granules that contain HMGA2 mRNA and other let-7 targets, including LIN28B, and to stabilize HMGA2 by opposing the action of let-7 on HMGA2 mRNA. This effect was not achieved by direct competition between IMP3 and let-7, but by providing HMGA2 with an IMP3-dependent, membrane-less cytoplasmic domain (i.e., IMP granules) that is devoid of RNA-induced silencing complex (RISC) pathway components [282].

## 8. Conclusions

Accumulating evidence indicates not only the importance of miRNAs as molecular players in tumorigenesis, but also their valuable diagnostic and therapeutic applications. The term “exo-miRNAs” has been coined to describe miRNAs that are stored in cellular exosomes and released into biological fluids, where they can be easily detected due to their high stability to provide more precise diagnoses and more personalized therapeutic strategies [283].

Among the HMGA-targeting miRNAs, the let-7 family members undoubtedly have the greatest clinical relevance, even though some of the data are contradictory and need further investigation [284]. Let-7 expression has been shown to have prognostic value in many tumours [53,70,200,201,285]. Moreover, the evaluation of let-7 family member expression in body fluids (plasma, serum, urine, and stool) has demonstrated utility in the early detection of tumours, as a prognostic tool [284] and as a chemoresistance predictor [205]. The downregulation of let-7a-5p in serum has been shown to predict lymph node metastasis and prognosis in colorectal cancer patients [286], whereas the RKIP/let-7 pathway metastasis signature has been demonstrated to predict a high risk of metastasis in breast cancer with higher accuracy than the individual genes [287].

An interesting application is in the monitoring of patients during therapy for cancers, such as lung cancer and acute promyelocytic leukaemia, since reduced let-7 plasma levels are considered to be an indicator of relapse [288].

On the other hand, the HMGA-targeting miRNAs are also attractive for their potential applications in cancer therapy, acting as drug targets or as off-the-shelf drugs.

Accordingly, some anticancer drugs have been shown to act through let-7 activation/induction, and their use has been proposed alone or in combination with other treatments. The pan-deacetylase inhibitor panobinostat, for example, affects liver cancer cell lines by inducing the let-7b-mediated downregulation of HMGA2 [289]. Similarly, phenformin has been shown to target the let-7/HMGA2 axis and has been proposed to be effective in combination with temozolomide against glioblastoma [290].

Even more fascinating is the potential administration of let-7 mimics as a next-generation treatment for cancer [291]. In this context, the administration of let-7g miRNA was shown to sensitize fluorouracil-resistant human hepatoma cells [292], whereas a locked nucleic acid-GapmeR (LNA-GapmeR) containing a let-7b mimic was shown to suppress tumour growth in vivo by regulating MYC expression in multiple myeloma and potentially in other MYC-dependent cancers [293].

In addition to let-7, which appears to be a dominant miRNA that is involved in controlling HMGA proteins, particularly HMGA2, a knotty miRNA network clearly controls the regulation of HMGA protein expression. Obviously, each miRNA has been separately studied, but it is reasonable to assume that the final effect on the modulation of HMGA protein expression is achieved by the cooperation of multiple miRNAs and it is not necessarily a simple additive effect. In addition, the inverse mechanism is intriguing, i.e., the possibility that HMGA transcripts act as ceRNAs. This aspect, together with the observation that miRNAs have multiple targets, suggests that small changes in one factor could lead to connected and amplified biological outcomes, but the quantitative prediction of these effects is currently not possible.

In conclusion, HMGA post-transcriptional regulation mechanisms appear to be much more complex than expected, and this field will certainly experience strong development in the future, with new miRNA and other ncRNA involved in HMGA regulation.

## Figures and Tables

**Figure 1 ijms-21-00717-f001:**
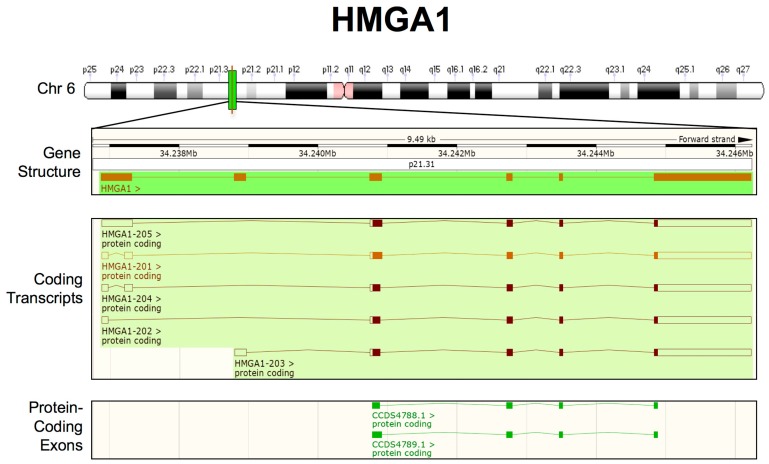
Updated human HMGA1 transcripts structure from Ensembl Genome Browser (GRCh38.p12-GCF_000001405.38). From top to bottom: chromosome 6 (Chr 6) labelled with chromosome bands (white, black and grey), centromeres (pink) and the *HMGA1* locus (green); the *HMGA1* gene structure, showing the entire length of the region considered, the location on Chr 6 (NC_000006.12):34, 236, 485–34, 246, 445, the chromosome band (p21.31) and HMGA1 exons/introns (orange boxes/lines); and the HMGA1 coding transcripts. The empty boxes indicate UTRs, whereas the filled boxes indicate open reading frames (ORFs). Merged Ensembl/Havana and Ensembl protein-coding data are shown in orange and brown, respectively; HMGA1 protein-coding exons are highlighted in green.

**Figure 2 ijms-21-00717-f002:**
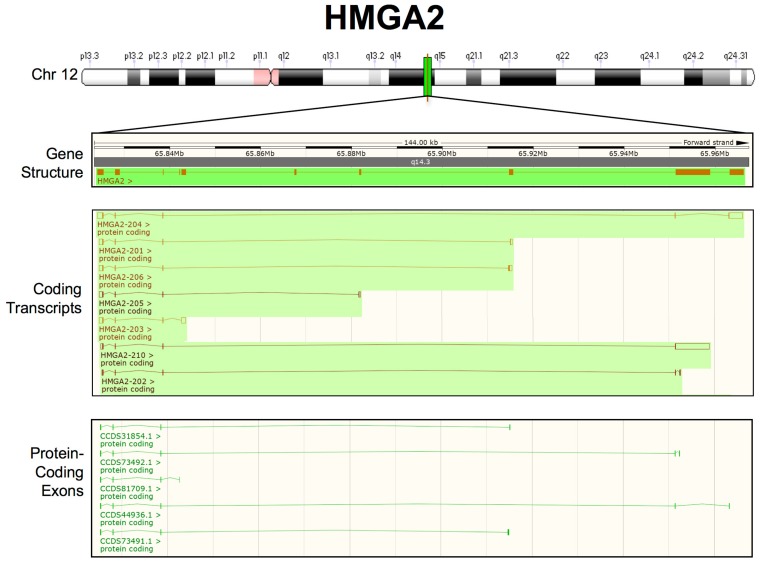
Updated human HMGA2 transcripts structure from the Ensembl Genome Browser (GRCh38.p13-GCF_000001405.39). From top to bottom: chromosome 12 (Chr 12) labelled with chromosome bands (white, black, and grey), centromeres (pink), and the *HMGA2* locus (green); the *HMGA2* gene structure, showing the overall length of the region considered, the location on Chr 12 (NC_000012.12):65,823,216–65,968,410, the chromosome band (q14.3) and HMGA2 exons/introns (orange boxes/lines); and, HMGA2 coding transcripts. The empty boxes indicate UTRs, whereas the filled boxes indicate ORFs. Merged Ensembl/Havana and Ensembl protein-coding data are shown in orange and brown, respectively; HMGA2 protein-coding exons are highlighted in green.

**Figure 3 ijms-21-00717-f003:**
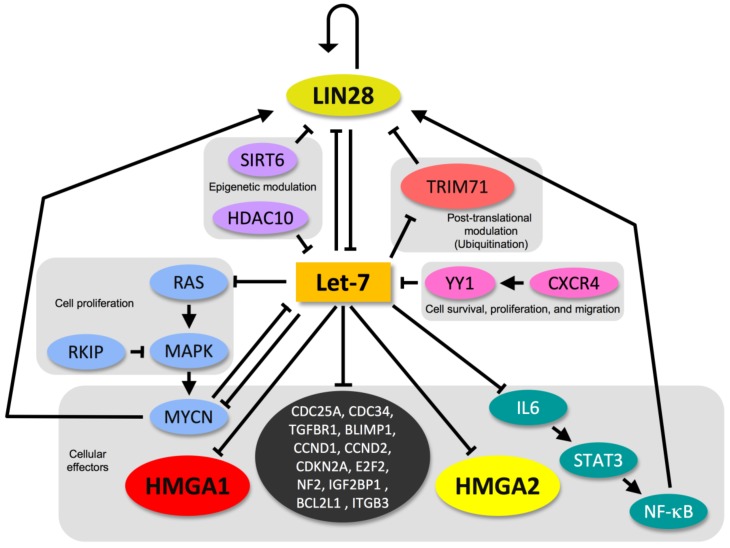
The LIN28/let-7 circuitry. HMGA proteins are inserted in the connected regulatory network of the LIN28/let-7 axis, which contains several regulatory feedback loops.

**Figure 4 ijms-21-00717-f004:**
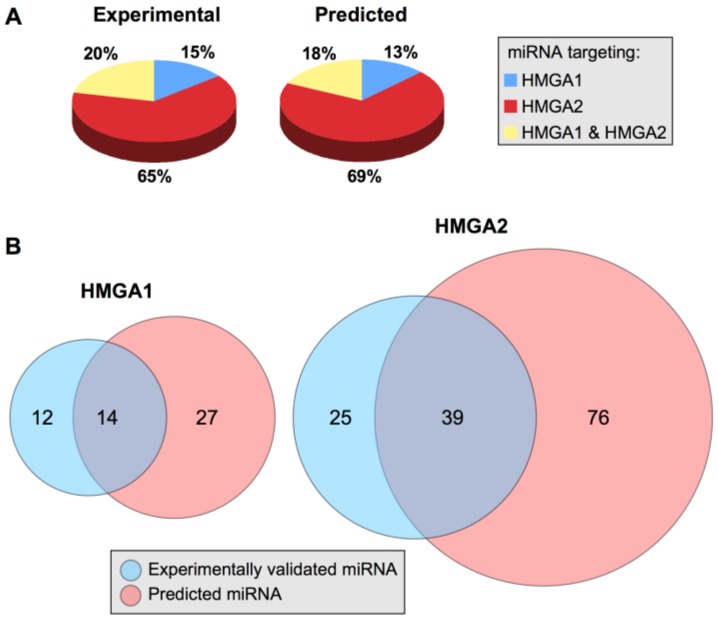
An overview of HMGA1- and HMGA2-targeting miRNAs. (**A**) Comparison of the percentage distributions of experimentally validated and predicted HMGA1- and HMGA2-targeting miRNAs. (**B**) Venn diagrams of experimentally validated and predicted miRNAs targeting HMGA1 and HMGA2.

**Table 1 ijms-21-00717-t001:** High mobility group A 1 (HMGA1)- and HMGA2-targeting of microRNAs (miRNAs).

miRNA	Exp.		TargetScan (CWCS)		Reference	miRNA	Exp.		TargetScan (CWCS)		Reference
	A1	A2	A1	A2			A1	A2	A1	A2	
let-7a-5p	✓	✓	−0.49	−2.67	[49,50,51,52]	302-3p				−0.15	
let-7b-5p	✓	✓	−0.49	−2.67	[45,53,54,55,56,57]	302a-5p			−0.03		[58]
let-7d-5p	✓	✓	−0.49	−2.67	[46,54,59,60,61]	325-3p				−0.05	
let-7e-5p		✓	−0.49	−2.67	[45]	326		✓		−0.08	[62]
let-7 g-5p		✓	−0.49	−2.67	[46,54,63,64]	329-3p				−0.23	
let-7c-5p		✓	−0.49	−2.67	[46,65,66,67]	330-3p		✓		−0.09	[68]
let-7a-5p		✓	−0.49	−2.67	[44,46,49,50,69,70,71,72,73]	330-5p		✓			[68]
let–7d–3p		✓			[74]	331-3p				−0.17	
let-7i-5p	✓				[75]	337-3p		✓			[76]
1-3p				−0.07		361-5p				−0.21	
7-5p			−0.08	−0.08		362-3p				−0.23	
9-5p		✓		−0.46	[77,78]	362-5p				−0.20	
10a-3p		✓			[79]	363-3p		✓		−0.15	[80,81]
15a-5p	✓	✓	−0.42	−0.55	[50]	365a-3p		✓		−0.26	[82]
16-5p	✓	✓	−0.42	−0.55	[50,83,84]	367-3p		✓		−0.15	[58]
17-5p				−0.32		369-3p				−0.08	
20-5p				−0.32		372-3p				−0.15	
21-5p		✓			[79]	373-3p				−0.15	
22-3p				−0.20		376-3p				−0.2	
23b-3p		✓		−0.17	[85,86]	376c-3p				−0.44	
25-3p				−0.15		379-5p				−0.27	
26a-5p	✓	✓	−0.60	−0.63	[50,87,88,89,90,91,92,93,94,95,96]	409-3p				−0.12	
28-3p				−0.18		410-3p				−0.06	
28-5p				−0.11		411-3p				−0.06	
32-5p				−0.15		411-5p.2				−0.14	
33a-5p		✓		−0.64	[97,98,99]	412-5p			−0.34		
33b-5p	✓	✓		−0.64	[80,100,101,102,103,104]	421				−0.22	
34a-5p		✓	−0.18		[105,106]	424-5p			−0.42	−0.55	
34b-3p	✓	✓			[62,106]	432-5p		✓			[62]
92-3p				−0.15		449-5p			−0.18		
93-5p				−0.32		452-5p				−0.14	
98-5p		✓	–0.49	−2.67	[107,108,109,110,111,112,113,114,115]	483-3p.1				−0.06	
101-3p		✓		−0.11	[116]	485-5p		✓			[117]
103–3p			−0.47	−0.32		486-5p			−0.05		
106a-5p		✓		−0.32	[118]	488-3p				−0.18	
107		✓	−0.47	−0.32	[119]	490-3p		✓		−0.24	[120,121]
124-3p	✓		−0.14		[122]	491-5p		✓	−0.24		[123]
125b-5p	✓	✓		−0.05	[124,125]	493-3p				−0.14	[126]
128-3p			−0.12	−0.14		493-5p				−0.1	
129-5p				−0.22		494-3p				−0.14	
132-3p				−0.32		495-3p		✓		−0.04	[127,128]
134-5p			−0.24			496.2				−0.16	
136-5p			−0.11	−0.16		497-5p	✓		−0.42	−0.55	[129]
137				−0.42		500b-5p				−0.20	
138-5p	✓		−0.21		[130]	503-5p		✓	−0.22	−0.31	[131]
140-3p.1			−0.05			505-3p.2				−0.03	
142-3p	✓	✓	−0.36	−0.72	[132,133,134]	506-3p			−0.14		
142-5p				−0.13		519-3p				−0.32	
143-3p			−0.07			520-3p				−0.15	
145-3p		✓			[135]	532-3p				−0.66	
146b-5p		✓			[136]	539-5p		✓			[137]
148-3p				−0.22		541-3p		✓			[138]
149-5p				−0.08		542-3p				−0.03	
150-5p		✓		−0.15	[139,140,141]	543		✓		−0.16	[142]
151-3p				−0.24		544a-5p			−0.08	−0.22	
152-3p				−0.22		548c-3p	✓	✓			[62]
154-5p		✓		−0.31	[143]	570-3p		✓			[62]
181-5p				−0.11		485-5p				−0.61	
182-5p				−0.20		491-5p				−0.31	
185-5p	✓	✓		−0.38	[144,145]	582-5p				−0.04	
186-5p		✓		−0.13	[146]	599		✓			[147]
190-5p				−0.34		625-5p	✓				[148]
194-5p				−0.18		653-5p				−0.09	
195-5p	✓	✓	−0.42	−0.55	[129,149,150,151,152]	655-3p				−0.15	
196a-5p	✓	✓	−0.42	−0.90	[50,153,154]	663a		✓			[155]
198	✓				[156]	664a-5p		✓			[157]
199-5p			−0.01			665			−0.37		
202-5p				−0.13		708-5p				−0.11	
204-3p		✓			[158]	758-3p	✓		−0.06		[159]
204-5p		✓		−0.44	[160,161,162]	760				−0.49	
206				−0.07		765	✓				[163]
211-5p		✓		−0.44	[164,165]	873-5p.2				−0.04	
212-3p				−0.32		892-3p				−0.14	
214-3p	✓				[166,167]	1224-5p				−0.01	
217			−0.26			1249-3p		✓			[168]
219a-5p		✓		−0.13	[169,170]	1297	✓	✓			[171,172]
221-3p		✓			[173]	1298				−0.04	
296-3p			−0.19	−0.15		3064-5p				−0.15	
296-5p	✓		−1.48		[174,175]	4458	✓				[176]
299-3p				−0.11		4500		✓			[177]

Exp.: experimentally validated HMGA-targeting miRNA. TargetScan (CWCS): HMGA-targeting miRNA predicted by TargetScan (miRNA families broadly conserved among vertebrates and miRNA families conserved only among mammals). The cumulative weighted context ++ score (CWCS) is provided for all predicted miRNAs. The names and sequences of experimentally predicted miRNAs were obtained from miRDB (http://mirdb.org/index.html). A1: HMGA1. A2: HMGA2.

**Table 2 ijms-21-00717-t002:** List of lncRNAs acting as ceRNA towards HMGA-targeting miRNA.

Name	Function	Role	Related Cancer	Reference
H19	Enhances HMGA2-mediated EMT by sponging let-7; upregulates HMGA1 expression by sponging miR-138	Oncogenic	Pancreas Colon cancer	[130,274]
CCAT1	Derepresses HMGA2 and c-Myc by acting as a molecular sponge for let-7	Oncogenic	Hepatocellular carcinoma	[275]
HMGA1P6 HMGA1P7	Acts as a decoy for HMGA1-targeting miRNAs such as miR-15, miR-16, miR-214 and miR-761	Oncogenic	Thyroid cancer Pituitary adenomas	[269,276]
NEAT1	Regulates the miR-211/HMGA2 axis	Oncogenic	Breast cancer	[164]
HULC	Increases HMGA2 expression by sequestering miR-186	Oncogenic	Liver cancer	[146]
RPSAP52	Enhances HMGA1 and HMGA2 protein expression in a ceRNA-dependent manner via miR-15a, miR-15b, and miR-16	Oncogenic	Pituitary adenomas	[48]
ANRIL	Improves cisplatin-sensitivity by regulating the let-7a-HMGA2 axis	Oncogenic	Ovarian cancer	[277]

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
