# Peer review of "High Mobility Group A (HMGA): Chromatin Nodes Controlled by a Knotty miRNA Network"

_ijms, 2020, doi:10.3390/ijms21030717_

Round 1
Reviewer 1 Report
The authors describe a miRNA network which regulates HMGA expression. This is a detailed review with clear illustrations and extremely easy to follow.
Some more description of details would improve this article.
Page 4, line 102: The authors mentioned that both HMGA1 and HMGA2 are generated by alternative splicing. Which transcripts in Figure 1 are responsible to these protein isoforms? The author may show the information in the text or a figure legend.
Page 8 line 223: The authors should change “adenocarcinoma” to “lung adenocarcinoma”.
Page 8 line 248: Typo; miR33b
Page 9 line 280: reference style should be arranged.
Page 12 line 390 and line 393: Adjust indents in these sentences.
Table 2: HCC should be spell out?
Author Response
We thank the reviewer for the positive comments.
Regarding the specific issues:
R: Page 4, line 102: The authors mentioned that both HMGA1 and HMGA2 are generated by alternative splicing. Which transcripts in Figure 1 are responsible to these protein isoforms? The author may show the information in the text or a figure legend.
A: We added the information in the text.
R: Page 8 line 223: The authors should change “adenocarcinoma” to “lung adenocarcinoma”.
A: We changed “adenocarcinoma” to “lung adenocarcinoma”.
R: Page 8 line 248: Typo; miR33b
A: The correction has been inserted in the text and others typo have been checked as well.
R: Page 9 line 280: reference style should be arranged.
A: The reference style has been arranged.
R: Page 12 line 390 and line 393: Adjust indents in these sentences.
A: The indents have been adjusted.
R: Table 2: HCC should be spell out?
A: HCC has been spelled in Hepatocellular carcinoma
Reviewer 2 Report
In this manuscript, Sgarra and co-workers present a timely review of the recent literature on the miRNA-mediated post-transcriptional regulation of the High Mobility group A proteins (HMGA1 and HMGA2). This manuscript is interesting and well presented, and I only have few minor comments to improve the quality of the review.
Page 4, line 123: the authors stated that the HMGA2 gene is more complex than HMGA1 “due to its length and the presence of a very large intron involved in rearrangements”. This seems to be a very vague sentence: what kind of rearrangements is this intron involved into?
Page 5, line 155: the authors stated that, in the present manuscript, they decided to discuss only few miRNAs involved in the regulation of HMGA, “those that we believe are the most relevant to HMGA expression modulation”. Were these miRNAs selected because they are the most widely studied? Or because they are specifically relevant to a particular field, i.e. cancer? I would suggest to provide a more precise statement on the rationale used for selecting this subset of miRNAs.
Table 1: I suggest to include in the top row the indication of what column corresponds to which HMGA (I assume left is HMGA1 and right HMGA2).
Page 15, line 533: in conclusion, the authors stated that “this field will certainly experience strong development in the future”. Again, this is a vague and not much meaningful sentence. It would be more interesting to have the authors’ perspective on what is the envisioned future direction that this development will take. Some considerations on what are the challenges associated with studying the HMGA-miRNAs axis would also be important to be included in the conclusion remarks.
Author Response
We thank the reviewer for the positive comments. Regarding the minor points:
R: Page 4, line 123: the authors stated that the HMGA2 gene is more complex than HMGA1 “due to its length and the presence of a very large intron involved in rearrangements”. This seems to be a very vague sentence: what kind of rearrangements is this intron involved into?
A: We have specified better the sentence and added two references. In the new version we indicate “ ……a very large intron involved in rearrangements leading to truncated or chimeric HMGA2 transcripts especially in benign mesenchymal tumours [184,185]
R: Page 5, line 155: the authors stated that, in the present manuscript, they decided to discuss only few miRNAs involved in the regulation of HMGA, “those that we believe are the most relevant to HMGA expression modulation”. Were these miRNAs selected because they are the most widely studied? Or because they are specifically relevant to a particular field, i.e. cancer? I would suggest to provide a more precise statement on the rationale used for selecting this subset of miRNAs.
A: They were selected because they are the most widely studied. We modified the sentence in the text accordingly.
R: Table 1: I suggest to include in the top row the indication of what column corresponds to which HMGA (I assume left is HMGA1 and right HMGA2).
A: In the table we indicated A1 and A2 for HMGA1 and HMGA2. In the new version this was explained in the legend to Table 1. (A1: HMGA1. A2: HMGA2.)
R: Page 15, line 533: in conclusion, the authors stated that “this field will certainly experience strong development in the future”. Again, this is a vague and not much meaningful sentence. It would be more interesting to have the authors’ perspective on what is the envisioned future direction that this development will take. Some considerations on what are the challenges associated with studying the HMGA-miRNAs axis would also be important to be included in the conclusion remarks.
A: We modified the conclusion as suggested by the reviewer.